# Protein Language Models Expose Viral Immune Mimicry

**DOI:** 10.3390/v17091199

**Published:** 2025-08-31

**Authors:** Dan Ofer, Michal Linial

**Affiliations:** Department of Biological Chemistry, Life Science Institute, Faculty of Science and Mathematics, The Hebrew University of Jerusalem, Jerusalem 91904, Israel; dan.ofer@mail.huji.ac.il

**Keywords:** adaptive immune system, autoimmune diseases, deep learning, epitope, feature selection, IL-10, immunological tolerance, PLM, ProteinBERT

## Abstract

Viruses have evolved sophisticated solutions to evade host immunity. One of the most pervasive strategies is molecular mimicry, whereby viruses imitate the molecular and biophysical features of their hosts. This mimicry poses significant challenges for immune recognition, therapeutic targeting, and vaccine development. In this study, we leverage pretrained protein language models (PLMs) to distinguish between viral and human proteins. Our model enables the identification and interpretation of viral proteins that most frequently elude classification. We characterize these by integrating PLMs with explainable models. Our approach achieves state-of-the-art performance with ROC-AUC of 99.7%. The 3.9% of misclassified sequences are signified by viral proteins with low immunogenicity. These errors disproportionately involve human-specific viral families associated with chronic infections and immune evasion, suggesting that both the immune system and machine learning models are confounded by overlapping biophysical signals. By coupling PLMs with explainable AI techniques, our work advances computational virology and offers mechanistic insights into viral immune escape. These findings carry implications for the rational design of vaccines, and improved strategies to counteract viral persistence and pathogenicity.

## 1. Introduction

The interplay between hosts and pathogens involves complex interactions, with viral evasion of the biological immune system being a crucial survival strategy. Viruses often employ molecular mimicry, adopting biophysical characteristics of their host, such as length and sequence composition, to evade immune detection. Understanding these mechanisms is essential for advancing therapeutic interventions, designing effective vaccines, and minimizing adverse immune reactions.

Viruses are classified (e.g., Baltimore classification; seven groups) based on how viral mRNA is produced and the nature of their genome (RNA, DNA, and strand properties) [1]. RNA viruses are typically distinguished from DNA viruses by their smaller genome size and fewer functional proteins [2]. Among the large dsDNA families, *Herpesviridae* infections are estimated to affect most of the human population. A defining feature of *Herpesviridae* is their capacity for latency. Specifically, after the initial infection, herpes simplex virus (HSV), varicella-zoster virus, human cytomegalovirus, Epstein–Barr virus, and others may enter a dormant phase [3]. These herpesviruses have developed mechanisms to evade host immune surveillance [4,5]. Human genomes also harbor endogenous retroviruses (ERVs), remnants of ancient infectious retroviruses [6]. While usually dormant, ERVs can be reactivated by various stimuli and in some cases may limit the spread of viral pathogens [7,8].

Viral and human proteins are expected to exhibit distinct features reflecting their divergent evolutionary origins [9]. While viral adaptation to hosts is evident at the level of codon usage, maximal optimization was assigned to genes encoding capsid and coat proteins [9]. Many clinically important viruses, such as human immunodeficiency virus (HIV) and influenza A (seasonal flu) [10], encode only a handful of proteins, essential for completing the viral life cycle. All viruses encode proteins that facilitate host entry, replication, assembly, and structural formation (e.g., envelope and capsid subunits) [11]. While these viral-specific features are largely absent from human proteins [12], hundreds of cases have been documented where human genetic material was co-opted by viruses during evolution, and vice versa [13]. Importantly, viruses have also evolved proteins dedicated to immune evasion, targeting both the innate and adaptive immune systems [14,15].

Viral mimicry highlights critical steps in establishing latency and manipulating the host’s immune response. For example, Epstein–Barr virus (EBV) is etiologically linked to Burkitt and Hodgkin lymphoma [16]. Epstein–Barr nuclear antigen 1 (EBNA1) shares linear and conformational similarity with host autoantigens, allowing the virus to manipulate host gene expression related to immune responses [17]. Structural similarity between EBNA1 and host proteins may also lead to cross-reactive antibodies, a hallmark of several autoimmune diseases [18]. Additional viral strategies exploit other host cellular machinery, including the priming of pathogenic responses [19]. Molecular mimicry has been extensively studied in the context of autoimmunity, focusing on how viruses imitate short segments of human proteins to escape immune recognition [20]. For example, analysis of 134 human-infecting viruses confirmed the widespread use of short linear amino acid mimics, particularly among Herpesviridae and Poxviridae [21]. EBV exhibits especially high levels of mimicry in proteins associated with autoantibodies in multiple sclerosis patients [22,23]. These findings suggest that viral mimicry not only contributes to immune evasion but also to the development of autoimmune diseases, highlighting potential therapeutic targets.

In recent years, there has been a rapid surge in the development and application of pretrained deep learning (DL) language models [24]. Pretraining refers to training models on large-scale unlabeled data to learn generalizable sequence patterns and representations. This unsupervised learning step equips models with robust prior knowledge, which can then be fine-tuned to enhance performance on specific downstream biological tasks. Biology has increasingly benefited from the rapid advancements in pretrained language models (PLMs). Modeling of biological sequences (DNA, RNA, and proteins) has achieved remarkable performance through DL techniques. Protein language models such as ProteinBERT [25], ProtGPT2 [26], ESM, and others have reached state-of-the-art results across diverse tasks, including protein structure and function prediction, mutational effect analysis [27], de novo protein design, post-translational modification prediction [28], and many more [29].

Despite these successes, it remains unclear whether protein language models encode representations that parallel (or diverge from) biological sequence-processing mechanisms. Analogous work in natural language and vision has shown that neural networks can learn biologically plausible low-level features, yet also rely on unexpected cues (e.g., texture over shape) [30,31]. Furthermore, their susceptibility to adversarial examples that mislead the model highlights important differences in robustness and perception between biological and DL-based systems [32,33,34].

In this study, we evaluate the ability of PLMs to differentiate viral proteins from their human hosts, with a particular focus on human–virus sequence pairs. We show that both PLMs and the natural immune system struggle to recognize viral proteins that closely mimic host sequences. Our findings provide new insights into the mechanisms by which viruses evade host immunity, revealing a striking correspondence between the classification errors of PLMs and the recognition failures of the host immune response. Understanding these parallel misclassifications may inform strategies to mitigate the impact of viral mimicry on human health.

## 2. Materials and Methods

### 2.1. Protein Datasets

All reviewed human proteins and virus proteins with a known vertebrate host were downloaded from SwissProt within the UniProtKB database (https://www.uniprot.org; 20 August 2023). We have restricted the database to status: reviewed, non-fragment (as of November 2021) and compressed to Uniref50 and Uniref90 sequence similarity clusters. Each protein sequence was paired with its annotations from the UniProt keywords, name, taxonomy, virus-host, and length.

Duplicate sequences at the UniRef90 level were dropped to reduce redundancy. Proteins longer than 1600 were excluded. The dataset was shuffled and partitioned by UniRef50 clusters into a training subset (80%) and a test subset (20%). All proteins sharing the same UniRef50 cluster (≥50% sequence identity) were assigned as one entity to either the training or test set. Thus, no members of the same cluster appear in both. Protein-level embeddings were downloaded from UniProt. Virus family, genus, and Baltimore classification were downloaded from ViralZone [35]. There are >200 families and 17k viruses and virions, prior to filtration.

### 2.2. Pretrained Deep Language Models (ESM, T5)

ESM2 is a deep learning architecture, based on the BERT Transformer model [36]. It was pretrained on the UniRef50 dataset to predict masked-out amino acids (tokens). It can efficiently represent amino acid sequences and has shown good performance across different protein predictive tasks. In this study, we have used different-sized ESM2 models. We use mean pooling for extracting a sequence-level representation of each protein. This approach has been shown to yield a good representation in numerous sequence-level problems. Specifically, the final dense layer of the chosen model is taken and its representation of each token (an individual amino acid) in the sequence is averaged over all tokens in the sequence. This representation can be followed by training on a specific task. We also downloaded pre-extracted protein-level embeddings, from UniProt (called T5). These embeddings were derived from “prottrans_T5_xl_u50”, a T5 PLM, with 3 billion parameters [29]. These can be used as input features for training downstream, non-deep ML models.

### 2.3. Human-Virus Model Training and Implementation

Models were trained to predict whether a protein belongs to a human or a virus. Performance was evaluated on the test set. The pretrained ESM2 models were fine-tuned in PyTorch (ver2.20) using the HuggingFace transformers library [37]. DL models were trained on a 16 GB, 4080 NVIDIA GPU with an 8-bit Adam optimizer, fixed learning rate 5 × 10^−4^, 16-bit mixed precision training, LoRA (low-rank adapters), batch size 16, max sequence length 1024, and cross-entropy loss for three epochs.

For finetuning the DL models, we used LoRA, a parameter-efficient fine-tuning method for transformers, implemented using PEFT (parameter-efficient fine-tuning) [38]. Briefly, during fine-tuning, LoRA replaces updates (Δ𝕎) to the original weight matrix (𝕎) with the decomposition of Δ𝕎 into two low-rank matrices *A* and *B*. Only the weights of *A*, *B* are updated during back-propagation. After training has been completed, *A* and *B* can be multiplied to yield Δ𝕎 and to update the underlying weight matrix. This approach uses ~1% as many trainable parameters as regular fine-tuning, allowing for faster training times, larger models, and has outperformed regular fine-tuning in some cases [38]. LoRA adapters were attached to all linear and attention layers, with LoRA rank *r* = 8, scaling factor *α* = 8 and no dropout.

The ESM models were trained using only sequences. Scikit-learn implementations and default hyperparameters were used for logistic regression (LR) and histogram gradient boosting decision tree (GBT) models [39]. The length baseline is a LR model trained only on sequence length. The amino acid (AA) n-grams model is a GBT trained on length and AA n-gram frequencies features [9].

### 2.4. Finding and Analyzing Model Mistakes

Following model training and evaluation, in order to analyze model errors and misclassifications, we performed an additional stage of extracting predictions over the whole dataset. We used the static embeddings model (“Linear-T5”). We rejoined the train and test data together and extracted predictions for the combined data, using four-fold, group-stratified cross-validation, with retraining in each split. Sequences were again partitioned by UniRef50 clusters. Finally, for each test split predictions that differed from the ground truth were marked as “mistakes”. Namely, if a model predicted a human protein for being a virus (abbreviated H4V), or vice-versa (V4H).

To understand the mistakes made by the human virus model, we ran separate error analysis models, where the target was defined as whether our original model had made a mistake. Multiple partitions of the data were analyzed separately: the dataset (25,117 sequences), a human-only subset (18,418 sequences), and only viruses from genera that are specific for humans as their host (3915 sequences). New features were extracted using the SparkBeyond autoML framework (see example in [40]). Inputs included the protein sequence, length, taxonomy, name, UniProt keywords, virus-host species, and Baltimore classification, but not embeddings. Features were automatically ranked by their support (i.e., number of examples), lift (i.e., the likelihood of a target class under the distribution induced by the feature, using an optimal binary split), and mutual information (MI) with the target, and selected for explanatory value [41].

### 2.5. Dimensionality Reduction of Features

t-SNE (t-distributed Stochastic Neighbor Embedding) dimensionality reduction was used to visualize the embeddings and map it to a low-dimensional space [42,43].

### 2.6. Model Performance

To assess the different model’s classification performance, we use the common metrics of AUC (area under the receiver-operating characteristic curve. AUC reflects the model’s discriminatory ability across thresholds and the trade-off between sensitivity and specificity, with an AUC of 0.5 reflecting a random predictor. Model performance metrics definition: Precision, recall and accuracy metrics are calculated as follows:(i)(Precision = TP/(TP + FP)(ii)Accuracy = (TP + TN)/(TP + TN + FP + FN)(iii)Recall = TP/(TP + FN)
where TP: true positive, FP: false positive, FN: false negative, and TN: true negative. The AUC and log-loss were calculated using scikit-learn. Precision and recall used macro averaging (arithmetic mean across all classes).

### 2.7. Immunogenicity Datasets and Scores

To assess immunogenicity, we used the IEDB Class-I immunogenicity predictor (http://tools.iedb.org; v2.27 release (25 May 2023) [44]. Although originally validated on 9-mer peptides, the toolkit allows predictions for the immunogenicity scores of peptides of any length. We randomly sampled 200 proteins for each combination of origin (human vs. virus) and classification outcome (correct vs. misclassified), for a total of 800 proteins.

## 3. Results

Our objective was to evaluate the performance of protein language models (PTMs) in distinguishing between human and viral proteins and to analyze the types of errors these models make. This aims to deepen our understanding of how computational models can mirror biological processes, particularly in the contexts of taxonomic classification and immune evasion.

### 3.1. Human Virus Models

Table 1 summarizes the models’ performance on the held-out test set. We evaluated multiple PLM model families and sizes, with larger models achieving the best results. While the amino acid (AA) n-gram model, using only sequence length and amino acid combinations, achieved good separation (91.9% AUC), the PLM-based models reached 99.7% AUC and ~97% accuracy. Training a logistic regression (“linear”) or tree model on top of the T5 embeddings (“T5”) was competitive with full fine-tuning, while requiring negligible computation and offering greater training stability and reproducibility.

### 3.2. Error Analysis Models Insights

In our analysis of errors made by the most stable human-virus model (Table 1, Linear-T5), we observed an overall mistake rate of 3.9% across the joint dataset. The complete list of proteins, features, and labels is provided in the Appendix A. Overall, the models more frequently misclassified viral proteins as human (V4H) than human proteins as viral (H4V). Specifically, 9.48% of viral proteins were misclassified (635/6699), compared to only 1.87% of human proteins (345/18,418; H4V), representing a five-fold difference. Note that immune-related annotations are not exclusive, but rather represent one of several enriched categories of misclassified proteins.

### 3.3. Virus Errors Analysis

Table 2 lists features associated with an elevated fraction of mistakes. We calculated the lift to determine the enrichment relative to a prior mistake background. The presence of endogenous retroviruses in the human genome results in a high rate of these H4V mistakes; these sequences are arguably not “human”. Endogenous retroviruses are of viral origin and have become embedded in the human genome. Short protein length and sparse keyword annotation also contribute to misclassification, along with other features (Table 2). Viral proteins annotated as involving the adaptive immune system are also extremely elusive, reflecting their evolved roles.

We found that mistakes are higher for genera of human-specific when compared to genera of viruses that infect vertebrates. Specifically, the 3915 viral proteins from genera with a human host (i.e., assumed coevolution related to human targeting viruses) are mistaken more (10.2%) than the overall rate for any vertebrate-host viruses, while the 2787 human-host viruses are even more confounding (11.6%). The assumption is that coevolution that led to adaptation underlies such trend in the model’s error rate.

### 3.4. Latent Structure Embeddings Clustering

We applied t-SNE to visualize the sequences’ embeddings (Figure 1A). While human and viral proteins cluster somewhat separately, the mistakes of the type V4H and H4V are quite distributed, and no sign of aggregation is observed. While the human-virus protein features and embeddings are different both in terms of sequence composition, length, and sequence embedding, they are not linearly separable, and mistakes are widely distributed throughout the latent space, as opposed to forming distinct clusters (e.g., of naturally occurring endogenous retroviruses).

Applying supervised UMAP as dimensionality reduction embedding (Figure 1B) resulted in a different partition, where both viral and human proteins display good separation to relatively purified clusters. Note that the viral and human-origin proteins are in a close proximity to each other. In addition, there are only 10s of proteins that are remote from each other, often forming small, isolated clusters that are viral or human-originated (marked by the different symbols, Figure 1B). We will not further discuss the nature of these isolated proteins.

We then tested whether the mistake rates associated with Baltimore class (Table 3) capture the mistakes within the viral families. Table 4 shows that there is a large difference in the mistake rate between major viral classes (e.g., class IV) and the specific viral families.

Inspecting the nature of the mistakes, we found that viruses that often cause long-term or life-long diseases are prominent among them. Prominent examples include Hepatitis E (liver disease), HIV (AIDS), HPV (Papilloma) and more. A full list of families and genera is provided in our repository. Appendix A provides a list of all human and viral sequences (total 25,117), model predictions and mistakes.

### 3.5. Immunogenicity Analysis

Immunogenicity scores reveal differences in how the immune system, mirrored by PLMs, is sensitive to host versus viral proteins. In Figure 2, we analyze the immune epitope database (IEDB) predicted immunogenicity score distributions across four distinct combinations: virus (Figure 2A), human proteins (Figure 2B), with (top) or without mistakes (bottom).

The IEDB immunogenicity scores reflect the propensity of a sequence to elicit an immune response, with higher scores indicating greater potential for detection by T-cells [44]. The observed distributions suggest differential immune recognition for viral versus human proteins, which can be essential for understanding viral escape mechanisms. As expected, viral proteins tend to have more extreme immunogenicity scores than human proteins. The low scores indicate epitopes that often go unnoticed. This supports the idea that viral proteins, especially those that imitate human proteins, have evolved under pressure from the immune system. As a result, they tend to fall at the extremes of immunogenicity; either they are highly detectable by the immune defenses (as threats that induces cell response) or they very well evade the immune surveillance disguised to avoid immune detection by alternative mechanisms (e.g., masked, glycan covered or otherwise camouflaged).

Notably, the subset of mistakes where the PLM mistakenly identified viruses as human (virus, mistakes, V4H; Figure 2A, top) demonstrates a tight clustering of scores around lower values. This indicates a subset of viral proteins that effectively mimic host immunogenicity patterns, blurring the lines for both biological and algorithmic detection.

Conversely, the true-positive viral detections (virus, no mistakes, Figure 2A, bottom) generally show a broader and higher range of scores, supporting the immune system’s ability to recognize and respond to these viral entities more robustly. For human proteins, PLM mistakes (Figure 2B, top) display a score distribution pattern that suggests a false flag of immunogenicity, possibly reflecting sequences with viral-like properties. They represent a plausible case for detecting the remnants of ancient viral infections or endogenous retro-elements within the human genome.

An intriguing observation is that irrespective of origin, the proteins falsely identified, whether viral or human (Figure 2 (top; H4V, V4H), share more commonalities in their immunogenicity profiles with each other (with mean immunogenicity scores around −0.5 to −0.55) than with their correctly categorized counterparts. This pattern unveils a potential blind spot in both biological and algorithmic recognition systems, suggesting that certain protein features associated with immune evasion are consistently challenging to discern. This insight sheds light on the intricacies of host–pathogen interactions and identifies an area for improving the accuracy of PLMs for numerous biomedical applications such as vaccine development and antiviral drug design.

### 3.6. V4H Mistakes Expose Traces of Host Sequences Within Viruses

We illustrate cases underlying immune escape mechanisms, specifically in the context of human proteome. While the human proteome (SwissProt, reviewed, see Methods) includes approximately 20k coding genes, viral proteomes are unstable. ViralZone [1] covers approximately 12.8k reference viral proteomes, with most viruses containing only a few proteins. In most instances, viral proteins are shorter relative to their representative hosts, excluding polyproteins such as gag-pol polyprotein. To avoid the complexity of multidomain proteins, we tested the nature of the PLM mistakes (V4H) by focusing on conserved domains [13].

Viral T-cell receptor beta chain-like (classified as ssRNA-RT; Feline leukemia virus; 321 aa) that was identified by misclassification (V4H, Figure 3). The predicted structure reveals two immunoglobulin-like (Ig) folds (Figure 3, 3D model). The main InterPro domains are the V-set (IPR013106) and C1-set (IPR003597). The V-set resembles antibody variable domains found in T-cell receptors (e.g., CD4, CD86), while the C1-set domain occurs almost exclusively in immune-related molecules, including Ig light and heavy chains, MHC class I and II complexes, and T-cell receptors.

Despite only moderate sequence identity (49%, Figure 3), the viral protein retains all the key features of the T-cell receptor beta, which normally functions in peptide–MHC recognition on antigen-presenting cells. The N-terminal domain of the viral protein adopts a V-set fold (Figure 3, top), spatially mimicking the variable domain of a TCR β chain. Although somewhat speculative, by presenting an epitope overlapping the peptide–MHC contact surface of a host TCR, such a mimic may compete with or divert antibody responses. This strategy would support viral immune escape by suppressing effector T-cell activation [46]. The integrated provirus that is present in hematopoietic stem cells persists in a transcriptionally silent state and remains invisible to cytotoxic T lymphocytes (CTLs). In conclusion, the presence of Ig-like domains in this viral protein underscores their role in cell–cell recognition and receptor mimicry, enabling the virus to compete with the host immune system.

### 3.7. Among the V4H Mistakes Are Proteins That Support Immune Escape Through Mimicry

Among the 65 reported V4H mistakes by our model (with high confidence in mistakenly predicting the viral sequence as a human protein, Appendix A), a protein named BCRF1 encodes interleukin-10 like protein from the Human herpesvirus 4 (EBV strain AG876; HHV-4). Figure 4A shows the taxonomical view of the interleukin-10 (Pfam domain of IL-10). IL-10 is an immunosuppressive cytokine produced by T cells, B cells, macrophages, and dendritic cells. The sequence is part of the UniRef50 cluster (P0C6Z6, 302 proteins) that is represented by IL-10-like proteins from a large number of organisms and diverse viruses.

Inspecting the taxonomical clock (Figure 4A) shows that viruses (red color) and eukaryotes (green) have many representatives that share a similar sequence, structure and function. From a functional perspective, the IL-10 is a key player in inhibiting pro-inflammatory cytokine induction in viruses such as Epstein–Barr virus (EBV), equine herpesvirus (EHV), and cytomegalovirus (CMV). The IL-10 is an example of a viral protein that acquired a sequence whose protein product can attenuate the host immune response. Interestingly, while not all 242 listed viral proteins infect humans, Parapoxvirus genus (belonging to the family of Poxviradae that includes Orf viruses (ORFV)), can lead to human disease indirectly through contact with an infected host (e.g., cattle), resulting in a shutdown of the human immune response via the viral IL-10 homolog.

Among the 65 cases on top V4H mistakes by the model (Appendix A), we discuss a few genes that directly relate to the ability of the virus to evade the immune system. Figure 4B demonstrates the Poxin–Schlafen genes of the Monkeypox virus (strain Zaire-96-I-16; MPXV). Specifically, this long viral protein (503 amino acids) carries a mechanism to overcome the interferon-based alarm system of infected cells. Poxins abrogate the human innate immune response by neutralization of the cGAS-STING axis. Upon detection of viral DNA in cell cytosol, cyclic GMP-AMP synthase (cGAS) synthesizes a second messenger (2′,3′-cGAMP) to induce type-I interferon via the adaptor STING. With the Poxin–Schlafen protein, the monkeypox virus gains an advantage for replication and dissemination in its host [47]. Interestingly, despite minimal sequence similarity, the viral Poxin and the host Schlafen fold (SLFN) share strong 3D similarity (Figure 4B), which supports the notion of immune-suppressive mimicry [48].

Another interesting example is the Surface glycoprotein CD59 homolog from the Saimiriine herpesvirus 2 (strain 11) (SaHV-2). CD59 in mammals operates as a complement-regulatory checkpoint. Multiple mammalian viruses exploit this function to evade innate immunity [49]. A number of enveloped viruses incorporate host-derived CD59 into their budding membranes. This acquisition confers a protection against membrane-attack complex (MAC)-mediated lysis. The similarity in structure and function of CD59 between the human gene and SaHV-2 expands to the GPI-anchor, the number and position of cysteine bridges, and the predicted fold. The PLM wrongly predicted the CD59-like gene from SaHV-2 as a human protein, the resemblance to host CD59 prolonging the viral replicative window [50]. We conclude that among the proteins that were indicated by mistaken classification (V4H), immunological functions, cell recognition, and adhesion (Appendix A) prevail.

## 4. Discussion

Our main finding is that PLMs can successfully distinguish human from viral proteins without sequence alignment or explicit similarity measures. We then focus on the remaining errors to bridge computational behavior with biological interpretation in the human–virus context. Inspecting the nature of the cases in which the computational model failed highlights instances where the human immune system also tends to fail in recognizing and eliminating those same families of virus pathogens. We find that the models’ errors are similar to those of the biological immune system. Notably, the misclassifications highlight protein families where the human immune system also tends to fail, suggesting an intriguing parallel between model errors and mechanisms of immune evasion. The failure to eradicate latent viruses has been proposed as a potential cause of many of the autoimmune diseases (AIDs) [51] and were also implicated in cancer research and immunotherapy [52].

For decades, the leading assumption was that AIDs initiated from missed recognition by the immune system of some pathogen’s epitope (often unknown) via mimicry [53,54,55]. Recent population-scale evidence strengthens the causal connection between EBV and multiple sclerosis [22]) and implicates additional AIDs (e.g., rheumatoid arthritis, systemic lupus erythematosus, consistent with γ-herpesvirus mimicry and related herpesviruses (e.g., HHV-6) [56,57].

It was shown that using an algorithm to improve protein sequence annotation by comparing amino acid-level embeddings improved conventional sequence similarity methods (e.g., BLAST) and pooled embedding models for homologous detection [58]. Our work takes an alternative approach; we use supervised PLMs and interpretable ML methods (e.g., decision-tree models with feature-based explanations) for this task, achieving state-of-the-art results (e.g., [59]), while often human endogenous retroviruses were associated with H4V and V4H mistakes (this study). Language models for viral escape were presented, where specific mutations altered the meaning while maintaining the grammaticality context. The analogy was applied to influenza, HIV, and SARS-CoV-2 proteins [60].

Lastly, DL embeddings and attention can be challenging to interpret, especially in a biological context [61]. To address this, we applied an autoML model that extracts explicit, human-readable features to explain the errors made by our DL model. This approach employs not just a different feature set, but also features which could not be used by any parent model, such as class labels, and label-specific partitions (e.g., Baltimore classes). By separating interpretability from prediction, we highlight which properties most strongly drive misclassification while ensuring model validity (and avoiding target leakage) (e.g., [40]).

Several of the reported functions shared by a viral and a human protein were identified only through structural relatedness (i.e., using SwissModel or AlphaFold2) and were not evident by sequence alignment (e.g., BLAST). An example is the viral protein DP71L (MyD116 homolog), which was mistakenly labeled by our model as a human protein. The protein is from the African swine fever virus and plays a role in the regulation of protein synthesis suppression during stress. It was shown that via mimicry, the protein forms a complex with host proteins to ensure precise dephosphorylation that secures translation. Such a short viral peptide (71 amino acids) also acts to inhibit the host interferon signaling pathway [62]. We recall that most mistaken predictions remain unresolved due to the limited knowledge associated with the uncharacterized viral proteins.

We illustrated instances of errors by the PLM model for proteins across a wide range of viruses (Figure 3 and Figure 4). IL-10 illustrates how herpesviruses have used a cytokine-like product to attenuate the immune response. Interestingly, mimicking as a general strategy was validated in CMV and other herpesviruses. Specifically, following infection, the VPS4 is recruited through the viral MIM2 motif, mimicking host ESCRT-III machinery. The mistakes of PLMs cover herpesviruses proteins that are structurally and functionally similar to host ESCRT-III components, highlighting a strategy for hijack cellular processes [63].

In our study, the PLM model can be used to identify viral–host overlooked pairs; many are involved in recognition and escape from the immune system. However, it should be noted that the complexity of immune recognition is far beyond the immunogenicity landscape. For example, personalized genetic predisposition and the nature of the HLA locus are critical to resolve the molecular arms race between host and viral proteins. Instead, we provide evidence that PLMs and taxon classification pretext tasks might serve as a proxy for studying and predicting immune evasion, potentially aiding in the prediction and design of vaccine candidates [60].

Inspecting the evolutionary strategies developed by the identified viruses to evade the host immune system may also have an impact on oncology in view of the advances in immunotherapy. The characterization of specific viral proteins that were misclassified could be extended to other pathogens. Specifically, it becomes evident that sequence and structural homology between tumor-associated antigens and peptides from microbiota species may influence tumor progression by affecting T cell responses [64]. Using PLMs to identify analog peptides with improved features from non-self-microbial antigens may be of great benefit for cancer vaccination and immunotherapy [52,65].

We assert that the PLM used in this study captured the grammar and semantics of the viral and host proteins. Moreover, many identified mistakes are similar to those occurring in nature, when viral proteins evade the immune system or interfere with a proper activation that elicits a productive response, reinforcing that the model’s residual errors are themselves biologically meaningful.

## 5. Conclusions

In conclusion, PLMs can effectively distinguish human from viral proteins, and the small set of misclassifications is highly informative. The errors of V4H and H4V are enriched in low-immunogenic, immune-evasive proteins and known viral mimicry factors (e.g., cytokine homologs, TCR variable region, C59 complement receptor). By combining PLMs with an interpretable error-analysis framework, we uncover sequence- and annotation-level features linked to misclassification, offering mechanistic insights into immune evasion and host–virus adaptive evolution. These findings suggest that PLM behavior partially mirrors host–virus recognition dynamics and may serve as a useful proxy for flagging candidate immune-evasive proteins.

## Figures and Tables

**Figure 1 viruses-17-01199-f001:**
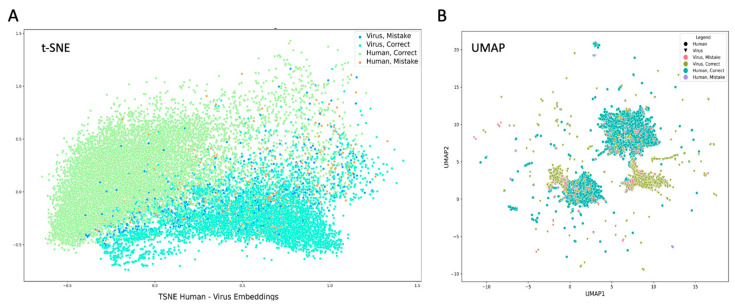
Human-virus embeddings. (**A**) t-SNE embeddings. (**B**) Supervised UMAP (Uniform manifold approximation and projection) embeddings, based on human/virus labels. Supervised UMAP uses the target label as part of the dimensionality reduction. All together there are 25,117 sequences, including 18,418 from the human proteome (see Appendix A).

**Figure 2 viruses-17-01199-f002:**
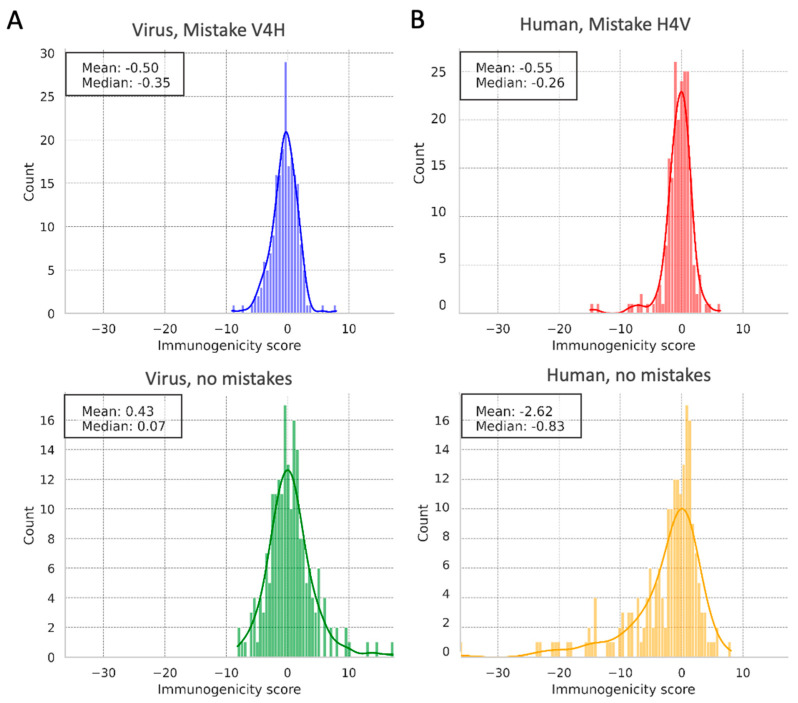
Distribution of the IEDB (immune epitope database) immunogenicity scores. (**A**) Viral proteins with and without mistakes (top and bottom, respectively). (**B**) Human proteins with and without mistakes (top and bottom, respectively). Note that the mean and median scores for the proteins that were mistaken by the DL model are substantially different. The relative immunogenicity score of predicted epitopes based on a two-sample Kolmogorov–Smirnov test is statistically significant, for virus (*p*-value < 0.001) and human samples (*p*-value < 1 × 10^−5^). The list of all protein representatives along with the results of the prediction models and marked as mistake of the type V4H and H4V is available in Appendix A.

**Figure 3 viruses-17-01199-f003:**
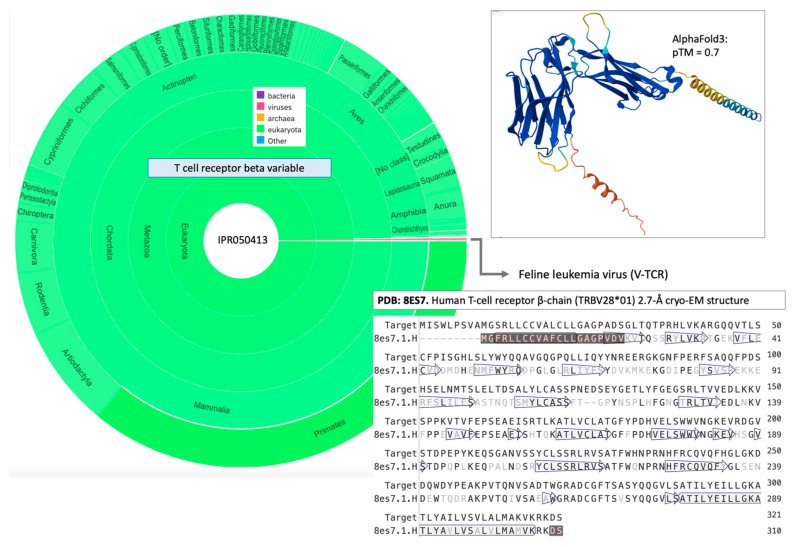
Taxonomical distribution of InterPro domain IPR050413 (T cell receptor beta variable, Ig superfamily). The pie chart shows that nearly all sequences belong to eukaryotic metazoans (~6000 proteins, green), with only a single viral sequence identified as misclassified (V4H). This viral gene, the V-TCR (321 aa) from Feline leukemia virus, belongs to the family Retroviridae and genus Gammaretrovirus. The predicted 3D fold (AlphaFold, pTM = 0.7) closely resembles mammalian T-cell receptor beta chains. Helices at the N- and C-termini correspond to the signal peptide and transmembrane domain, respectively. Homology modeling with SwissModel [45] confirmed close similarity to mammalian TCR beta chains. In particular, sequence alignment to the human TCR β-chain (resolved by cryo-EM at >2.7 Å; PDB: 8ES7), which recognizes a cancer-testis antigen [46], shows moderate sequence identity (49%). Faded letters indicate sequence differences. Secondary structure elements are annotated along the sequence, with α-helices marked as rectangles and β-sheets as arrows.

**Figure 4 viruses-17-01199-f004:**
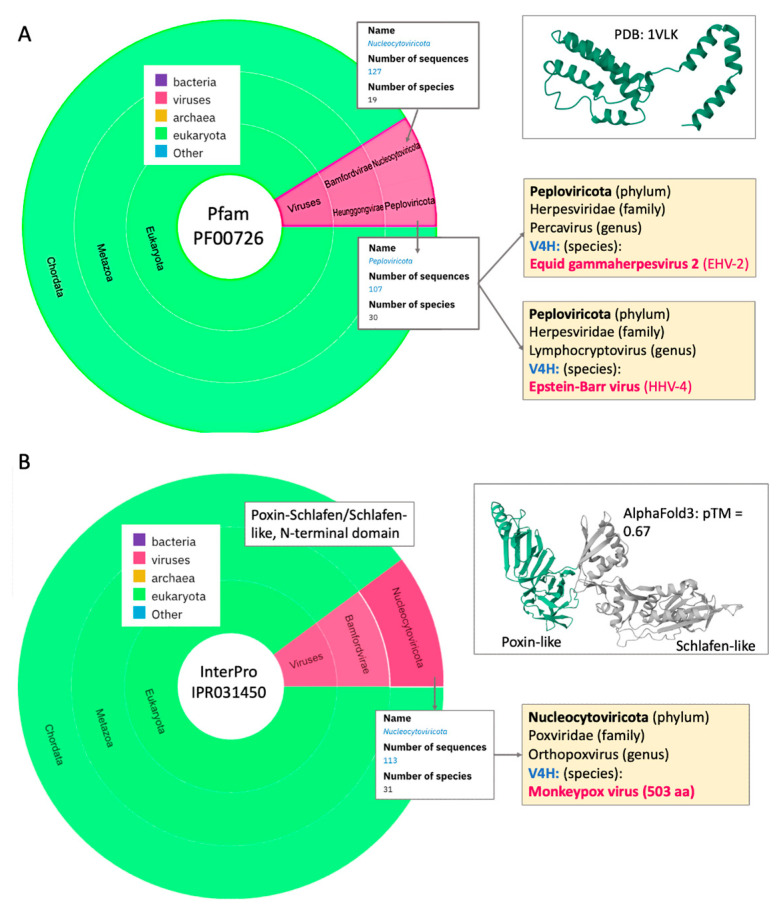
Taxonomical view for cases of V4H mistakes involved in immune escape. (**A**) Analysis of the IL-10 domain (Pfam: PF00726). A dominant occurrence in Biletaria (2k proteins, ~90%, green) and viruses (~10%, red). There are 234 such viral proteins (red). The section of the viruses is split into the family Herpesviridae (107 proteins) including many viruses that infect humans. In the yellow frames are examples of V4H mistakes of the model for EHV-2 and HHV-4. A 3D-structure of a viral representative is shown (PDB: 1VLK) and is conserved across the vial and human proteins with the IL-10 domain. (**B**) Analysis of the InterPro domain (IPR031450) with 113 sequences (from 31 species; red color). Among these viruses, many viruses infect humans. The example of V4H mistakes of the model is of a major viral group belonging to the family Poxviridae including smallpox. In the yellow frames is the example of Monkeypox virus. The structure prediction from AlphaFold 3 is shown with high confidence to the mammalian/human homologs. The viral protein contains poxin-like and Shlafen domains.

**Table 1 viruses-17-01199-t001:** Human-virus classification models performance.

Model ^a^	AUC (%)	Accur.	Prec.	Recall	Log-Loss
BL Length	61.97	78.5	78.5	78.5	0.52
AA n-grams	91.5	88.49	88.49	88.49	0.28
ESM2 8M	98.09	94.72	92.15	92.33	0.2
ESM2 35M	98.69	95.83	93.81	93.92	0.18
ESM2 150M	99.26	96.99	95.54	95.48	0.12
ESM2 650M	**99.67**	**97.86**	**96.85**	96.68	0.09
Linear-T5	99.56	97.57	97.57	97.57	**0.06**
Tree-T5	99.65	97.7	97.7	**97.7**	**0.06**

^a^ BL, baseline; AA, amino acids; T5, T5 model embeddings from UniProt. Values are in %. The best performance results across the modes are in bold. Accur. Accuracy; Prec. Precision.

**Table 2 viruses-17-01199-t002:** Overall features of mistaken proteins.

Features	Mistake Rate (%)	Number of Proteins	Lift ^a^
“Adaptive immune” KW	60.5	46	15.5
Endogenous retrovirus	30	40	7.7
Oncogene KW	19.3	393	4.9
Sequence length <170	12.1	4539	3.1
Virus	9.4	6699	2.4
Name “putative”	8.7	1050	2.2
Few KW (<8)	8.8	3326	2.2

^a^ Lift is the ratio of the frequency of mistakes relative to the overall baseline. KW, keyword.

**Table 3 viruses-17-01199-t003:** Mistakes by Baltimore class.

Baltimore Class	Genome	# of Families	^a^ Rep. Species	Mistake Rate (%)	# of Proteins	^b^ Lift
VII	dsDNA-RT	1	HBV-C	34.2	108	3.6
VI	ssRNA-RT	1	FeLV	19.5	666	2
II	ssDNA	3	HPV B19	13.1	129	1.3
I	dsDNA	13	HHV-4	8.2	4421	0.8
IV, V	ssRNA	28	VSIV	8.1	1017	0.8
III	dsRNA	4	RV-B	0.8	358	0.1

^a^ Representative species among the misclassified families. The full names and taxonomical details are based on ViralZone and available in Appendix A. ^b^ Lift is calculated relative to prior mistake rate of viruses (9.47%). #, number

**Table 4 viruses-17-01199-t004:** Mistakes by virus family (V4H).

Viral Family	Class	^a^ Main Disease	Mistake Rate (%)	Support	^b^ Lift
Hepeviridae	IV	Hepatitis	44.4	9	4.7
Hepadnaviridae	VII	Hepatitis	34.3	108	3.6
Circoviridae	II	CNS infection	33.3	27	3.5
Polyomaviridae	I	Cancer	30.7	62	3.2
Picornaviridae	IV	Nose/Throat	28.6	7	3.0
Retroviridae	VI	Cancer/AIDS	19.5	666	2.1
Polydnaviriformidae	I	N.A.	18.4	49	1.9
Arteriviridae	IV	N.A.	18.2	22	1.9
Papillomaviridae	I	Cancer	14.2	520	1.5
Caliciviridae	IV	Intestines	13.8	29	1.5
Paramyxoviridae	V	Mumps	12.9	124	1.4
Anelloviridae	II	Immune Supp.	11.5	52	1.2

^a^ Representative disease by viruses. N.A. No human-specific viral family or genera. ^b^ Lift is relative to the prior mistake rate of viruses (9.47%, as in Table 3).

## Data Availability

Additional information code and data are available in: https://github.com/ddofer/ProteinHumVir/.

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
