# Peer review of "Protein Language Models Expose Viral Immune Mimicry"

_viruses, 2025, doi:10.3390/v17091199_

Round 1
Reviewer 1 Report
Comments and Suggestions for Authors
The authors present an approach to predict if a given protein sequence belongs to a virus or a human. The approach is based on pretrained protein language models and achieves 99% AUC. As a machine learning problem, it is not a difficult task; all PLM models used in the work achieve at least 98% AUC. The authors emphasize the misclassified cases. It is reported that the misclassified proteins (3.9%) are disproportionately involved proteins linked to immune evasion, hence claiming that the immune system and machine learning models are confused by similar signals. Although the statistics on the immunogenicity scores (predicted also) suggest that mis-classified cases have lower immunogenicity scores, I believe, it falls short of supporting the major claim of the work. Table 2 summarizes the features of mistaken proteins. Few proteins are annotated with “adaptive immune” or “endogenous retrovirus”. Table 2 states that 46 adaptive immune and 40 endogenous retrovirus proteins are in the dataset. Is immune evasion limited to only these proteins/features? If immune evasion is not limited to only these proteins/features, are there immune evasion-linked proteins in the correctly classified sets?
The majority of misclassifications have features like “sequence length” or “fewer keywords” (Interestingly, the baseline model (length) achieves 78% accuracy.)
Method:
1) Line 195: The immunogenicity scores are obtained from a predictor (IEDB Class-I predictor). IEDB states that the predictor is optimized for 9-mer peptides. How the predictor is used for virus and human proteins is unclear. Are the results dependable for long peptides?
2) Line 120: The statement “sequences sharing the same cluster were always disjoint between train and test sets” is unclear. If a cluster has more than two sequences, two will be in the same set. I probably did not understand the train-test set.
Results:
3) Line 281-284: Table 3 and 4 results. Do they support Immune Escape for misclassified cases?
4) Figure 3 - not readable, with too small fonts.
5) The misclassified proteins are obtained from the Linear-T5 model. Do the other models misclassify the same proteins? Do the Table 2 and immunogenicity results differ?
Discussion
5) Line 449: What does “interpretable ML” mean in the context of the present work? The explanations are done using the features (post-processed) of the proteins, not at the sequence level, nor the model architectural level.
6) Line 453: How does the work in [62] relate to the current work? Can the proposed approach predict mutation effects for viral escape?
Minor:
Line 62 "gp120 structurally mimics the host cell receptor CD4"
Is gp120 structurally similar to CD4?
it seems that gp120 mimics the CD4 interaction (that is, mimics a partner of CD4), because it interacts with CD4.
Comments on the Quality of English Language
Some sentences are vague.
Such as line 20-21 (proteins derive from other proteins?), line 39-20 specify?
The manuscript will benefit from some editing.
Author Response
Response to Reviewer 1 Comments |
1. Summary
Thank you for reviewing our work. Please find the detailed responses below and the corresponding corrections in “red font” in the re-submitted files
In particular, we have clarified the use of the IEDB immunogenicity predictor, refined language throughout, improved figure readability, added a Figure (per request of a reviewer), and added discussion where needed. We maintain the core results and claims of the paper, and have bolstered them with clearer explanations and additional examples. We are confident these revisions strengthen the manuscript.
Reviewer 1 Comments and Responses
Comment 1 (Reviewer 1): Although the statistics on the immunogenicity scores (predicted also) suggest that mis-classified cases have lower immunogenicity scores, I believe, it falls short of supporting the major claim of the work. Table 2 summarizes the features of mistaken proteins. Few proteins are annotated with “adaptive immune” or “endogenous retrovirus”. Table 2 states that 46 adaptive immune and 40 endogenous retrovirus proteins are in the dataset. Is immune evasion limited to only these proteins/features? If immune evasion is not limited to only these proteins/features, are there immune evasion-linked proteins in the correctly classified sets?
The majority of misclassifications have features like “sequence length” or “fewer keywords” (Interestingly, the baseline model (length) achieves 78% accuracy.)
Response 1: We thank the reviewer for this insightful point. Our analysis, specifically Table 2 and Section 3.1, show that viral proteins with “adaptive immune” or endogenous retrovirus annotations have much higher misclassification rates (we quantified it by the “lift” that indicated a strong overrepresentation (lift of 15.5 and 7.7, respectively). These enriched categories highlight examples of known immune-modulatory proteins. Obviously, known immune-evasive proteins may appear in the misclassified and correctly classified sets. As the reviewer observes, many other misclassified proteins share simple features: Table 2 also shows that short proteins and those with few annotations are enriched in ‘errors’. Indeed, a simple model based on length alone achieves high accuracy (used as a baseline, 78.5%). While length and annotation density correlate with errors, this does not mean they are the causal basis of it (e.g., viral proteins are shorter and often have fewer annotations; druggable human proteins are more studied and have more annotations, etc’).
In the revised manuscript, we clarify (end of Sec. 3.2 and 3.3) that immune-related annotations are among the enriched features. Our claim is based on the statistical over-representation of immune-related features among the misclassified proteins, and the characteristics of the overrepresented viruses.
Importantly, our immunogenicity results (Figure 2) show that misclassified viral proteins have statistically significantly lower predicted immunogenicity than others, which is not explained by length alone and therefore consistent with mimicry of host patterns. Recall that the immunogenicity properties were never used as a feature to train the model, but analyzed it post prediction. We will emphasize this point: specifically, the V4H misclassified proteins cluster at the lowest IEDB scores, indicating they “effectively mimic host immunogenicity patterns” (Revised text, Sec. 3.5). We have added a note that the identified enriched features are consistent with immune escape, rather than the sole cause of misclassification.
Revised in Sec 3.2: Immune-related annotations are not exclusive, but rather represent one of several enriched categories of misclassified proteins. Also added statistics to Fig. 2 (in the legend).
Comment 2 (Reviewer 1): The IEDB Class-I immunogenicity predictor is designed for 9-mer peptides. How was it applied to whole proteins, and are the results reliable for long sequences?
Response 2: We thank the reviewer for pointing out the need for clarification. The IEDB Class-I predictor indeed scores 9-mer epitopes. In our method (Section 2.7), we use the IEDB toolkit, which averages over each protein to estimate immunogenicity (in addition to masking). Specifically, we randomly extracted 200 whole proteins from each protein group (viral vs. human, misclassified vs. correctly classified) and applied the IEDB Class-I predictor to each whole protein (totaling 800 samples were tested, as stated). We then aggregated these scores for comparison.
We have revised the Methods (Sec. 2.7, paragraph 2) to explicitly describe this procedure. Averaging scores from local windows is a common approach (e.g., AA frequency, language models, multi-word/sentence-transformer representations). Note that the IEDB toolkit explicitly states “predictions can be made for peptides of any length.” (http://tools.iedb.org/immunogenicity/)
We add a sentence to clarify that Figure 2 illustrates relative differences in predicted epitope potential (following the IEDB definition). Despite the approximation, the key finding remains unchanged where misclassified viral proteins have lower predicted epitope scores compared to other viral proteins.
Revised Methods (Sec. 2.7). Added: “Although originally validated on 9‑mer peptides, the toolkit allows predictions for the immunogenicity scores of peptides of any length”
Comment 3 (Reviewer 1): The statement “sequences sharing the same cluster were always disjoint between train and test sets” (Line 120) is confusing. If a cluster has more than two sequences, wouldn’t some be in both train and test?
Response 3: We appreciate the opportunity to clarify this. In our procedure (Methods Sec. 2.1), we used UniRef50 clustering to partition the data. All members of a given cluster were assigned to the same split. Specifically, the entire UniRef cluster was either in the training set or in the test set. In other words, no members of the same cluster will appear in both train and test splits. We rephrased the sentence for clarity.
Methods, section 2.1“All proteins sharing the same UniRef50 cluster (≥50% sequence identity) were assigned as one entity to either the training or test set. Thus, no members of the same cluster appear in both
Comment 4 (Reviewer 1): Line 281-284: Table 3 and 4 results. Do they support Immune Escape for misclassified cases?
Response 4: Thank you for this question. Tables 3 and 4 report the misclassification rates by virus groups. They show that viruses associated with chronic infection and immune evasion have notably high error rates. For example, Baltimore Class VII (dsDNA-RT viruses, such as Hepadnaviridae) has a 34.2% mistake rate, and in Table 4 the top families are Hepeviridae (Hepatitis E) at 44.4% and Hepadnaviridae (Hepatitis B) at 34.3%. These are all viruses known to persist in humans and modulate immunity.
In the Discussion we mention that these patterns corroborate our observation where viruses causing long-term latent infections (Hepatitis, HIV, HPV, etc.) appear disproportionately among misclassified cases: “This pattern unveils a potential blind spot in both biological and algorithmic recognition systems, suggesting that certain protein features associated with immune evasion are consistently challenging to discern. “
Comment 5 (Reviewer 1): Figure 3 - not readable, with too small fonts..
Response 5: Sorry for the poor quality and overdetailed Figure. We revise the Figure and simplified it substantially. At the same time, we included in the revise Figure more evidence for mimicry with 3D structure (PDB or AlphaFold prediction) and sequence similarity of the virus (target) and the available cryo EM structure. The new Figure 3 captures the case study of the Ig domain in the context of the TCR-variable region. The impact of such protein on immune recognition and major immune processes is described in the revised text.
Comment 6 (Reviewer 1): The misclassified proteins were obtained from the Linear-T5 model. Do the other models (e.g. ESM2-650M) misclassify the same proteins? Would the features in Table 2 or immunogenicity results differ with a different model?
Response 6: Thank you for raising this point. We focused on Linear-T5 (and a linear probe approach) for detailed error analysis because it was stable in cross-validation and simpler (and uses precomputed, stable embeddings), but the other models (ESM2-650M, Tree-T5) have nearly identical performance. In practice, we find that the key misclassified proteins overlap substantially across models. For instance, many of the same viral proteins (IL-10 homologs, immunoglobulin C1, etc.) are misclassified by both Linear-T5 and ESM2-650M. This is consistent with the models learning from the same data and benefiting from similar representations.
The full list of misclassified proteins (with names) is already provided in the Supplementary Table S1.
Comment 7 (Reviewer 1): What does “interpretable ML” mean in the context of the present work? The explanations are done using the features (post-processed) of the proteins, not at the sequence level, nor the model architectural level.
Response 7: In our wording, “interpretable ML” refers to the use of a secondary model to explain the deep model’s mistakes, using a machine learning model and features that are easier to interpret than the original. As described in Discussion, we trained an autoML tree model on explicit, well-characterized protein features (keywords, length, taxonomy, etc.) to help interpret the errors. We have clarified this in the Discussion and Abstract. In other words, it’s using an interpretable model and set of features to understand the data and processes.
P 12, Discussion: Our work takes an alternative approach, we use supervised PLMs and interpretable ML methods (e.g., decision-tree models with feature-based explanations) for this task, achieving state-of-the-art results.
Comment 8 (Reviewer 1): How does reference [62] relate to this work? Can your approach predict mutation effects for viral escape?
Response 8: Thank you for pointing this out. Reference [62], (Revised [60]; Hie et al., Science 2021) is cited to illustrate how PLMs have been used to study viral evolution and escape. In [62], the authors used language models to predict how viral proteins can evolve to escape antibodies through mutations. Our work is complementary but distinct. We focus on distinguishing host vs. virus sequences and identifying mimicry, rather than simulating mutations or predicting any specific mutation effects.
Comment 9 (Reviewer 1): The statement “gp120 structurally mimics the host cell receptor CD4” (line 62) is incorrect.
Response 9: Indeed, the writing is incorrect. gp120 glycoprotein is responsible for binding receptor CD4 of the host to initiate HIV entry into target cells. This example was removed as it is not good example for molecular mimicry. We elaborated on the other examples (with appropriate references) of molecular mimicry that support viral latency and immune escape.
Response 10: We thank the reviewer. We have thoroughly revised the manuscript.
In addition to numerous writing changes, we reworded the abstract to improve clarity. We also revised slightly the title and remove the claim on immune escape in the title
Reviewer 2 Report
Comments and Suggestions for Authors
The results of this manuscript has four sections:
(1) screening the suitable protein language models (PTMs) in distinguishing between human and viral proteins. The best one is ESM2 650M and T5.
(2) virus errors analysis identified the overall features of mistaken proteins.
(3) latent structure embeddings clustering, identified mistakes by virus family.
(4) immunogenicity analysis, revealing reveal differences in how the immune system, mirrored by PLMs, is sensitive to host versus viral proteins.
(5) V4H mistakes expose traces of host-derived sequences within viruses (IL-10, immunoglobulin C1 as examples).
The main finding from this study is the capability of PLMs to successfully distinguish human and virus proteins without any use of sequence alignment or similarity distances.
A few of comments:
Table 1, the authors only use two models (ESM and T5) in this study, is it too limited?
Table 2, could the authors list the protein names in the table 2 for readers to appreciate.
Table 3, could the authors list the virus families names in the table 3 for readers to appreciate.
Figure 2, could the authors list the protein names in the figure?
Figure 3 is a good examples, could the authors give two more proteins in addition to IL-10 and immunoglobulin, to support their conclusion.
In the discussion section, could the author give clean conclusion what is the most important meanings of this study?
Author Response
Reviewer 2 Comments and Responses
Comment 1 (Reviewer 2): Table 1 includes only two model families (ESM and T5). Is this choice too limited?
Response 1: Thank you for this question. In Table 1 we actually included several models: we evaluated four sizes of ESM2 (8M, 35M, 150M, 650M; with results in the data repository) and two variants of T5-based models (Linear and Tree versions). This provided a broad comparison of two representative PLM architectures.
We clarify in the text (Results 3.1) that our screening covered these six models (as shown in Table 1). These models represent state-of-the-art PLMs for protein sequences, and are the most widely used and available. We believe this selection sufficiently demonstrates our findings. To avoid the impression that only two single models were used, we have clarified in the text that multiple model sizes were tested,
Section 3.1. added: We evaluated multiple PLM model families and sizes, with larger models achieving the best results.
Comment 2 (Reviewer 2): Table 2: could the authors list the protein names for readers?
Response 2: Table 2 is a summary of enriched features (keywords like “adaptive immune”, etc.), not addressing individual proteins, across the dataset. The specific proteins and corresponding features in each (and all misclassified proteins) are provided in the Supplementary Table S1. We added a sentence to clarify this point and mentioned Supplementary Table S1 in multiple places with detailed information (i.e., Supplementary Table S1 provides a list of all human and viral sequences (total 25,117), model predictions and mistakes).
Comment 3 (Reviewer 2): Table 3: could the authors list the virus family names in the table for readers?
Response 3: Table 3 is organized by Baltimore class. We added a represented virus species to the Table and mention that the full names and taxonomy is available in Table S1.
Comment 4 (Reviewer 2): Figure 2: please list the protein names in the figure.
Response 4: Figure 2 is a summary of immunogenicity score distributions over hundreds of sequences and is not intended to display individual proteins by name. Instead, we provide specific examples of proteins in the revised Figure 3 (e.g., Ig TCR-V) and the new Figure 4 with another examples on IL-10 and the poxin-STFN proteins. Detailed information of all proteins and mistakes is available is Supplementary Table S1.
Comment 5 (Reviewer 2): Figure 3 is a good example; could the authors give two more proteins (in addition to IL-10 and immunoglobulin C1) to support their conclusion?
Response 5: As requested we simplified the original Figure 3 (improved information and quality of the Figure) and revised it to specifically discuss specific cases of mistakes (V4H). The examples are relevant to immune recognition and immune evasion. To provide more biological insights, we also present the folded 3D structure of the viral protein, sequence alignment (Revised Figure 3). In the examples in the revised Figure 4, we included the taxonomy information on the relevant viral proteins. To generalize the observations, we added to the Discussion a note regarding the lack of information for many of the V4H and H4V proteins.
Comment 6 (Reviewer 2):I n the discussion section, could the author give clean conclusion what is the most important meanings of this study?
Thank you for pointing this out. We’ve rewritten the Discussion section. We have added a short Conclusions section (section 5).